# Interspecific Competitions between *Frankliniella intonsa* and *Frankliniella occidentalis* on Fresh Lentil Bean Pods and Pepper Plants

**DOI:** 10.3390/insects14010001

**Published:** 2022-12-20

**Authors:** Chun-Hong Yang, Feng-Jiao Qiao, Zhaozhi Lu, Chang-You Li, Tong-Xian Liu, Yu-Lin Gao, Bin Zhang

**Affiliations:** 1Key Laboratory of Integrated Crop Pest Management of Shandong Province, College of Plant Health and Medicine, Qingdao Agricultural University, Qingdao 266109, China; 2State Key Laboratory for Biology of Plant Diseases and Insect Pests, Institute of Plant Protection, Chinese Academy of Agricultural Sciences, Beijing 100193, China

**Keywords:** flower thrips, interspecific competitions, invasive species, phytophagous insects, species dominance

## Abstract

**Simple Summary:**

In this article, we studied the interactions between the invasive thrips pest *Frankliniella occidentalis* (WFT) and a native thrips *Frankliniella intonsa* (IFT) in China, where they co-exist on crop plants and WFT seems to outcompete IFT. In laboratory bioassays we focused on the reproductive mode of the thrips (sexual or parthenogenic), and the presence of a sugar source (honey or flowers), both variables that change over time during pest colonisation of a crop. Our results suggest that in the early stages of *F. occidentalis* invasion, the provision of honey (flowering) promotes this pest’s population development. This research will inform effective strategies for the management of *F. occidentalis* invasion and help reduce subsequent crop damages.

**Abstract:**

Background: Flower thrips (*Frankliniella intonsa*, IFT) and west flower thrips (*Frankliniella occidentalis*, WFT) are often found together on the host plant in China. WFT is an important invasive species that seems to outcompete the native IFT. In order to clarify the interspecific competitions between the two thrips, this study measured the population development of IFT and WFT under sexual and parthenogenetic reproductive modes on two hosts (fresh lentil bean pods with/without honey and pepper plants at seedling/flowering stages) in the laboratory. Results: When reared on fresh lentil bean pods (with/without honey), WFT population size was lower in mixed species populations compared to single species populations but the presence of WFT had nor negative effect on IFT population size. These results were dependent of the reproductive mode. When honey was supplied, the ratio of female-to-male in the progeny of WFT produced under sexual reproductive mode increased significantly in the presence of IFT. On pepper seedlings, mixed populations were more favorable to the population development of IFT at the seedling stage, but more favorable to WFT at the flowering stage. Conclusions: In the early stage of WFT invasion and colonization, the emergence of flowering and honey (nectar) sources may have a positive effect on the population development of WFT.

## 1. Introduction

*Frankliniella occidentalis* (Thysanoptera: Thripidae) is an important invasive pest on vegetables and flowers [1]. Since its discovery in the United States [2], *F. occidentalis* has expanded rapidly throughout the world [3] and is now distributed in many countries and regions [4]. *Frankliniella occidentalis* was first reported and became established in mainland China in 2003 [5]. Subsequently, *F. occidentalis* was observed in many places in China, and at present, damage caused by *F. occidentalis* has been reported in more than 10 provinces, autonomous regions, and municipalities [1,6]. *Frankliniella occidentalis* has a wide host range of more than 250 types of vegetables and flowers [7,8,9].

*Frankliniella intonsa* (Thysanoptera: Thripidae) is a native species and widely distributed in China. *Frankliniella intonsa* and *F. occidentalis* have similar morphology and habits and produce similar damage to their hosts [10]. *Frankliniella intonsa* and *F. occidentalis* usually occur together on the same plants and have overlapping ecological niches [11,12]. Both *F. intonsa* and *F. occidentalis* have a short developmental period, high fecundity, and both sexual and parthenogenetic reproductive modes. They are small, disperse easily, and feed in cryptic sites [13,14]. They can cause injury to their host plants by direct feeding or laying eggs [15,16,17]. *Frankliniella intonsa* and *F. occidentalis* damage the leaves of plants [18,19] and feed on the flowers [20,21] and fruits [22], affecting both plant photosynthesis [23] and fruit quality [7,24]. At the same time, these two species of thrips can spread a variety of plant viruses to their hosts causing more serious injury [25,26,27,28] and huge economic losses every year [29,30,31].

Some studies have shown that *F. occidentalis*, as an invasive pest, was more adaptable than *F. intonsa.* The survival rate of *F. occidentalis* was greater than that of *F. intonsa* under thermal stress [32]. The population growth of *F. occidentalis* was larger than that of *F. intonsa* when populations coexisted [10], and the population growth of *F. occidentalis* was larger than that of *F. intonsa* under imidacloprid stress [33]. Recent studies showed that the degree of injury caused by *F. occidentalis* varies greatly in different areas and that *F. intonsa* is the dominant species in some areas [34]. The reasons behind this variation are unknown. Both *F. intonsa* and *F. occidentalis* are capable of sexual and parthenogenetic reproduction. In parthenogenetic reproduction, females do not need to mate to reproduce as unfertilized eggs develop into haploid males [13]. When both male and female thrips are present, they will produce diploid females from fertilized eggs and haploid males from unfertilized eggs [14]. A female-biased sex ratio was initially produced by all mated females [35]. This phenomenon complicates the competition between the two thrips. In addition, the flowering period of the host plant has a great influence on the population dynamics of thrips. Both species tended to aggregate within flowers when these were available [36] and the population growth increased almost geometrically after flowering [37]. The population of thrips on vegetables increased during the flowering period and peaked in the full flowering period [38]. Does flower nectar affect competition outcomes? The purpose of this study was to explore the effects of the emergence of honey/nectar sources and different reproductive modes on the competitions between the two thrips.

## 2. Materials and Methods

### 2.1. Plants and Insects

Pepper (*Capsicum annuum,* Diwang336, Shouguang, Shandong, China) seedlings were grown in soil in 12-hole trays (540 mm × 280 mm × 58 mm). When three true leaves had developed, individual pepper seedlings were transplanted into a flowerpot (150 mm diameter and 132 mm high). The competitions experiment used pepper seedlings at two stages: the seedling stage (from four true leaves to the end of the seedling stage) and the flowering/fruit-setting stage. The duration of each stage was 28 d. Prior to sowing pepper seeds we added 200 g/m^2^ organic fertilizer consisting of 15 g/m^2^ urea, 30 g/m^2^ diammonium phosphate and 30 g/m^2^ potassium sulfate. At the pod setting stage, we added organic fertilizer consisting of 22.5 g/m^2^, urea is 7.5 g/m^2^, and potassium sulfate 7.5 g/m^2^. Lentil bean pods were picked when approximately 15 cm long and used for rearing thrips.

*Frankliniella occidentalis* were collected from clover on the campus of Qingdao Agricultural University (N 36°31′, E 120°39′). *Frankliniella intonsa* were collected from Hainan pepper flowers. Thrips were cultured on fresh lentil beans in an incubator at 28 ± 1 °C, 14 L:10 D photoperiod, and 50% ± 5% RH. Parthenogenetic rearing of *F. occidentalis* and *F. intonsa* was achieved by culturing single larvae in a 50 mL centrifuge tube with fresh lentil bean pods as food. For sexual rearing, about 100 larvae of *F. occidentalis* or *F. intonsa* larvae of the same age were placed in a glass jar (15 cm × 15 cm × 12 cm) that contained fresh lentil bean pods. All tested adults were randomly selected from these populations.

### 2.2. Experimental Design

#### 2.2.1. The Culture of Thrips under Sexual Reproductive Mode on Fresh Lentil Bean Pods

Ten males and 20 females of *F. occidentalis* or *F. intonsa* were cultured together in a glass jar (15 cm × 15 cm × 12 cm) as the non-competitions (control) groups. The non-competitions treatment for *F. occidentalis* and *F. intonsa* replicated 4 times. Five males and 10 females of *F. occidentalis* were cultured together with 5 males and 10 females of *F. intonsa* in a single glass jar as the competitions group and was this was replicated 4 times. The foods for all groups were beans soaked in 10% honey (as a nectar substitute) for 2 h for the honey treatment, or beans that had not been soaked in honey for the without honey treatment. Both treatments with and without honey were replicated 4 times. The food was replaced with fresh food each day until the female died. The progeny was raised individually to adults, and the gender and number were recorded (Table 1, No 1).

#### 2.2.2. The Culture of Thrips under Parthenogenetic Reproductive Mode on Fresh Lentil Bean Pods

In this case, 20 females of *F. occidentalis* or *F. intonsa* were cultured separately in glass jar (15 cm × 15 cm × 12 cm) as non-competitions (control) groups, and ten females of *F. occidentalis* and 10 females of *F. intonsa* were cultured together as the competitions group. The culture of each group was replicated three times. The foods for all groups were beans soaked in 10% honey (as a nectar substitute) for 2 h for the honey treatment, or beans that had not been soaked in honey for the without honey treatment. Both treatments with and without honey were replicated three times. The food was taken out and placed in a new glass tube every day until the maternal female died. The progeny was raised individually to adults, and the gender and number were recorded (Table 1, No 2).

#### 2.2.3. The Culture of Thrips under Sexual Reproductive Mode on Pepper Plants

Peppers at seedling stage or at the beginning of flowering stage and of similar health were placed individually in a plastic flowerpot (150 mm diam., 132 mm high). The flowerpot mouth was sealed with plastic wrap and sealing paper to prevent thrips from falling onto the ground. Newly emerged thrips that had not laid eggs were transferred from the test glass jars onto the peppers. Three groups were established: sexual *F. occidentalis* (20 females and 10 males of *F. occidentalis* were placed together), sexual *F. intonsa* (20 females and 10 males of *F. intonsa* were placed together) and mixture of sexual *F. occidentalis* and *F. intonsa* (10 females and 5 males each of *F. occidentalis* and *F. intonsa* were placed together). Each group was replicated three times. The thrips in the flowerpot were observed daily. After the experiment, a trematode tube was used to remove all the thrips and peppers. Next, they were treated with 70% alcohol and the number of females and males of each species were counted under a microscope (Table 1, No 3).

#### 2.2.4. The Culture of Thrips under Parthenogenetic Reproductive Mode on Pepper Plants

The same experimental set-up and method was used as above in 2.2.3, except the three groups were: parthenogenetic *F. occidentalis* (30 females of *F. occidentalis* were placed onto peppers), parthenogenetic *F. intonsa* (30 females of *F. intonsa* were placed onto peppers) and a mixture of parthenogenetic *F. occidentalis* and *F. intonsa* (15 females each of parthenogenetic *F. occidentalis* and *F. intonsa* were placed together onto peppers). Each group had three replicates. The thrips in the flowerpot were observed daily (Table 1, No 4).

### 2.3. Data Analysis

Growth index was calculated as:

Growth index = number of F/number of P,

Where P = parental generation and F = filial generation.

All data were analyzed by SAS 8.1 software, and the difference between various parameters with and without competitions was analyzed by the one-way analysis of variance (ANOVA).

## 3. Results

### 3.1. Effect of Interspecific Competitions on the Population Development of F. occidentalis and F. intonsa Cultured under Sexual Reproduction Mode on Fresh Lentil Bean Pods

When thrips were fed on fresh lentil bean pods, interspecific competitions did not affect the population growth index of *F. intonsa*, but the population growth index of *F. occidentalis* was significantly affected. Regardless of the presence or absence of honey, the population growth index of *F. occidentalis* decreased significantly after it was mixed with *F. intonsa* (Figure 1A,B). In the presence of honey, the female-to-male ratio in the progeny of *F. occidentalis* increased significantly after mixing with *F. intonsa* (Figure 2).

As a single species (non-competitions group), the population growth index of *F. occidentalis* was significantly greater than that of *F. intonsa* regardless of the presence (*F*_11,36_ = 14.98, *p* < 0.001) or absence of honey (*F*_11,36_ = 27.84, *p* < 0.001). When males and females were analyzed separately, the population growth index of *F. occidentalis* males (*F*_7,24_ = 14.41, *p* < 0.001) and females (*F*_7,24_ = 12.73, *p* < 0.001) was also significantly greater than that of *F. intonsa*. In the competitions (mixed species) group, the population growth index of *F. occidentalis* was no significant difference with that of *F. intonsa* (*F*_7,24_ = 20.25, *p* < 0.001). When males and females were analyzed separately, the population growth index of *F. occidentalis* males and females was also no significant difference with that of *F. intonsa*. (Figure 1A,B).

The population growth index of *F. occidentalis* was significantly affected by interspecific competitions. The population growth index of *F. occidentalis* in the competitions group was significantly lower than that in the non-competitions group. When males and females were analyzed separately, the population growth index of *F. occidentalis* females and males in the competitions group was also significantly lower than that in the non-competitions group. However, there was no significant difference in the population growth index of *F. intonsa* between competitions group and non-competitions groups regardless of the presence or absence of honey. When males and females were analyzed separately, there was also no significant difference.

In the absence of honey, the female-to-male ratio in the progeny of *F. intonsa* was significantly greater than that in *F. occidentalis* progeny in both groups (*F*_7,24_ = 1.57, *p* = 0.1916; Figure 2). In the presence of honey, female-to-male ratio in the progeny of *F. intonsa* was significantly greater than that of *F. occidentalis* under non-competitions conditions. However, in the competitions group, the female-to-male ratio in the progeny of *F. occidentalis* was not significantly different from that in the progeny of *F. intonsa*. In the presence of honey, the female-to-male ratio in the progeny of *F. occidentalis* in the competitions group was significantly greater than that in the non-competitions group, but there was no significant difference in female-to-male ratio in the progeny of *F. intonsa* between non-competitions and competitions groups.

### 3.2. Effect of Interspecific Competitions on the Population Development of F. occidentalis and F. intonsa on Fresh Lentil Bean Pods under the Parthenogenetic Reproductive Mode

Under non-competitions conditions and parthenogenetic reproductive mode, the population growth index of *F. occidentalis* was significantly greater than that of *F. intonsa* regardless of the presence or absence of honey (*F*_7,16_ = 44.38, *p* < 0.001; Figure 1C). In the competitions group, the population growth index of *F. occidentalis* was again significantly greater than that of *F. intonsa*. Under the parthenogenetic reproductive mode, the population growth index of *F. intonsa* in the non-competitions group was not significantly different from that in the competitions group regardless of the presence or absence of honey. However, the population growth index of *F. occidentalis* in the competitions group was significantly lower than that in the non-competitions group.

### 3.3. Effect of Interspecific Competitions on the Population Development of F. occidentalis and F. intonsa Cultured under Sexual Reproductive Mode on Pepper Seedlings

Under the non-competitions condition, the population growth index of *F. occidentalis* was significantly greater than that of *F. intonsa* when cultured at the seedling stage. In the competitions group, although the population growth index of *F. occidentalis* was significantly greater than that of *F. intonsa*, it was not significantly different from that in the non-competitions group. In contrast, the population growth index of *F. intonsa* in the competitions group was still significantly greater than that in the non-competitions group (*F*_3,8_ = 148.98, *p* < 0.001; Figure 3A).

When cultured at the flowering stage under non-competitions conditions, there was no significant difference in the population growth index between *F. intonsa* and *F. occidentalis*. In the competitions group, the population growth index of *F. occidentalis* was significantly greater than that of *F. intonsa*. The population growth index of *F. occidentalis* in the competitions group was significantly greater than that in the non-competitions group. However, the population growth index *F. intonsa* in the competitions group was significantly lower than that in the non-competitions group (*F*_3,8_ = 9.11, *p* = 0.0058; Figure 3B).

### 3.4. Effect of Interspecific Competitions on the Population Development of F. occidentalis and F. intonsa on Pepper Seedlings under Parthenogenetic Reproductive Mode

Following culture at the seedling stage under non-competitions conditions, the population growth index of *F. occidentalis* was significantly greater than that of *F. intonsa*. In the competitions group, there was no significant difference between the population growth index of *F. occidentalis* and that of *F. intonsa*. The population growth index of *F. occidentalis* in the competitions group was significantly lower than that in the non-competitions group. The population growth index of *F. intonsa* in the competitions group was not significantly different from that in the non-competitions group (*F*_3,12_ = 11.33, *p* = 0.0008; Figure 3C).

Following culture at the flowering stage under non-competitions conditions, the population growth index of *F. intonsa* was significantly greater than that of *F. occidentalis*. In the competitions group, the population growth index of *F. occidentalis* was significantly greater than of *F. intonsa*. The population growth index of *F. occidentalis* in the competitions group was not significantly different from that in the non-competitions group. The population growth index of *F. intonsa* in the competitions group was significantly lower than that in the non-competitions group (*F*_3,8_ = 22.57, *p* = 0.0003; Figure 3D).

## 4. Discussion

*Frankliniella occidentalis* is a highly invasive pest that has spread from its original range to a worldwide distribution. Superior competitive ability is considered one of the main reasons for *F. occidentalis*’ success in the invasion. However, in some areas, *F. occidentalis* is not the dominant species, the invasive *F. occidentalis* appears to be competitively excluded by the native thrips. For example, *F. occidentalis* occurs in low numbers in the eastern states of the USA, where the native *F. tritici* dominates [39,40]. Various biotic and abiotic factors affect the relative abundance and the direction of species displacement between *F. occidentalis* and locally present thrips species [34,41]. Our study showed that interspecific competition was influenced by the host plants, reproduction mode as well as nutrition (honey supply). Feeding on different plants or different stages of the same plant had different effects on the population growth index of *F. occidentalis* and *F. intonsa*. In the early stage of *F. occidentalis* invasion and colonization, the availability of nectar on flowering plants is likely to promote the population growth rate of *F. occidentalis*.

Some studies indicated that the relative abundances between *F. occidentalis* and native thrips have been determined by their host plant species differences [34]. *F. intonsa* and *F. occidentalis* both are highly polyphagous and frequently co-occurs on various vegetable plants such as kidney bean and pepper [42,43]. On kidney beans, the density of *F. intonsa* was significantly higher than *F. occidentalis* [17,44]. *F. occidentalis* was dominant over *F. intonsa* on pepper [38,45]. In this study, competition results depended on the host plants. When thrips received fresh lentil pods as food source, competition reduced the growth index of *F. occidentalis* but not that of *F. intonsa*. On pepper seedlings, competition increased the population growth index of *F. intonsa*, while on flowering peppers, the population growth index of *F. occidentalis* increased

When the host plants were fresh lentil bean pods, interspecific competition did not affect the population growth index of *F. intonsa*, but the population growth index of *F. occidentalis* was significantly affected. Regardless of the presence or absence of honey, the population growth index of *F. occidentalis* decreased significantly after it was mixed with *F. intonsa* under both reproductive modes, indicating that interspecific competition was detrimental to the population growth of *F. occidentalis*, which is consistent with the results from previous studies [34].

When fed on pepper seedlings, there was no significant difference between the growth index of *F. occidentalis* under sexual reproductive mode with and without mixing with *F. intonsa*. However, when the two thrip species were mixed under the parthenogenetic reproductive mode, the population growth index of *F. occidentalis* increased significantly, while the growth index of *F. intonsa* decreased significantly. When the two thrips were mixed under the sexual reproductive mode, the growth index of *F. intonsa* increased significantly. These results indicated that during the early stage of *F. occidentalis* invasion, pepper seedlings are poor host plants. During the later stage of *F. occidentalis* invasion, *F. intonsa* exhibited greater ability to adapt to interspecific competitions.

*F. occidentalis* is flower-dwelling thrips. The population of thrips increased during the flowering period and peaked in the full flowering period [46]. Nutrients in different host plants and flowers have a great impact on the growth and development of thrips [47]. Some studies believe that the proportion of soluble protein and soluble sugar in plants affects the host selection behavior of phytophagous pests. When the content of soluble protein is high, the survival rate, growth and development rate and fertility of pests are relatively improved [48,49]. Soluble sugar is an important nutrient and energy source, which can stimulate insect feeding and promote the oviposition of some species of thrips [50]. In our study, honey supply increases the female-to-male ratio of the progeny of *F. occidentalis* under sexual reproductive mode cultured on fresh lentil bean pods, and flower emergence increases the growth index of *F. occidentalis* under sexual reproduction mode.

In our study the presence or absence of honey had no significant effect on the population growth index of the two species of thrips. However, in the presence of honey, the female-to-male ratio in the progeny of *F. occidentalis* increased significantly after mixing with *F. intonsa* and under sexual reproductive mode. *F. occidentalis* has a typical haploid/diploid sex determination system. The increases in the proportion of female progeny indicate that, in the presence of honey, interspecific competitions promoted the mating efficiency of female *F. occidentalis*. Studies have shown that *F. intonsa* had a stronger ability to guard honey and fed longer on honey compared to *F. occidentalis* [34], which would reduce the activity space, thereby increasing the probability of mating and mating efficiency of *F. occidentalis*. This may explain the increased female-to-male ratio in the progeny of *F. occidentalis* after it was mixed with *F. intonsa* and honey is supplied.

When fed on pepper seedlings at the flowering stage, the population growth index of *F. intonsa* declined significantly in both reproductive modes, while the population growth index of *F. occidentalis* increased significantly in the sexual reproductive mode, and did not change significantly in the parthenogenetic reproduction mode. These results indicated that the flowering period is important for the promoting the population growth rate and development of *F. occidentalis* in the early stage of its invasion and colonization. This also may explain the dominance of *F. occidentalis* populations in areas close to flower markets.

Both parthenogenetic and sexual reproductive modes were used in this study to simulate competition between the two thrip species in both the initial invasion and subsequent colonization phases. The results showed that there was no significant difference in the population growth index of the two reproductive modes when thrips were cultivated on fresh lentil bean pods. Feeding on fresh lentil bean pods in the presence of competition was detrimental to the population development of *F. occidentalis*, but had no effect on the population development of *F. intonsa*. On pepper seedlings, the population growth index of *F. occidentalis* was greater than that of *F. intonsa* in the presence of competition.

Under non-competitions conditions (single species culture) on pepper seedlings, the population growth index of *F. occidentalis* was significantly greater than that of *F. intonsa* in both sexual and parthenogenetic reproductive modes. Under competitions conditions, although the growth index of *F. occidentalis* was still greater than that of *F. intonsa* under sexual reproduction mode, the growth index of *F. intonsa* was significantly increased compared to that under non-competitions conditions. Under the parthenogenetic reproductive mode, there was no significant difference in the growth index between the two kinds of thrips, indicating that interspecific competitions at the seedling stage were beneficial to the growth of *F. intonsa*, which is inconsistent with previous results [43]. One of the possible reasons is that the initial population used by Gai et al. [43] were adults that have been mating for three days, while this study used unmated adults. Therefore, after populations were mixed in this study, it not only involves the mixing at the oviposition period but also affects the mating process. When male and female thrips coexist, interspecific competitions have a positive effect on the mating and oviposition of *F. intonsa*. While under the parthenogenetic reproductive mode, interspecific competitions have an obvious negative effect on the growth of *F. occidentalis*. Further studies are needed to investigate the underlying reasons.

Under non-competitions conditions on pepper seedlings at flowering stage, the population growth index of *F. intonsa* was significantly greater than (under parthenogenetic reproductive mode) or equal to (under sexual reproductive mode) that of *F. occidentalis*, indicating that flowering period was more favorable to the growth of *F. intonsa*. However, under competitions conditions, the population growth index of *F. intonsa* decreased significantly under both reproductive modes, while the population growth index of *F. occidentalis* increased significantly under the sexual reproductive mode, indicating that *F. occidentalis* interfered with the population development of *F. intonsa*. We found that after the pepper blooms, most of the thrips that occupied the flowers were *F. intonsa* while *F. occidentalis* were mainly active on the leaves. This is consistent with previous results showing that *F. intonsa* had stronger ability to guard honey compared to *F. occidentalis* [51]. However, such occupation and possession of honey cost more energy, thereby reducing the population growth of *F. intonsa*.

In conclusion, the outcomes of interspecific competition between the two species of thrips depends on the host species, additional sugar sources, and reproductive mode. The results of this study showed that when fed on fresh lentil bean pods, the interspecific competitions between the two thrips significantly interferes with the population development of *F. occidentalis*. However, when fed on pepper seedlings, interspecific competition is more beneficial for the population development of *F. intonsa* during the seedling stage, while the flowering stage is more favorable for the population development of *F. occidentalis*.

## Figures and Tables

**Figure 1 insects-14-00001-f001:**
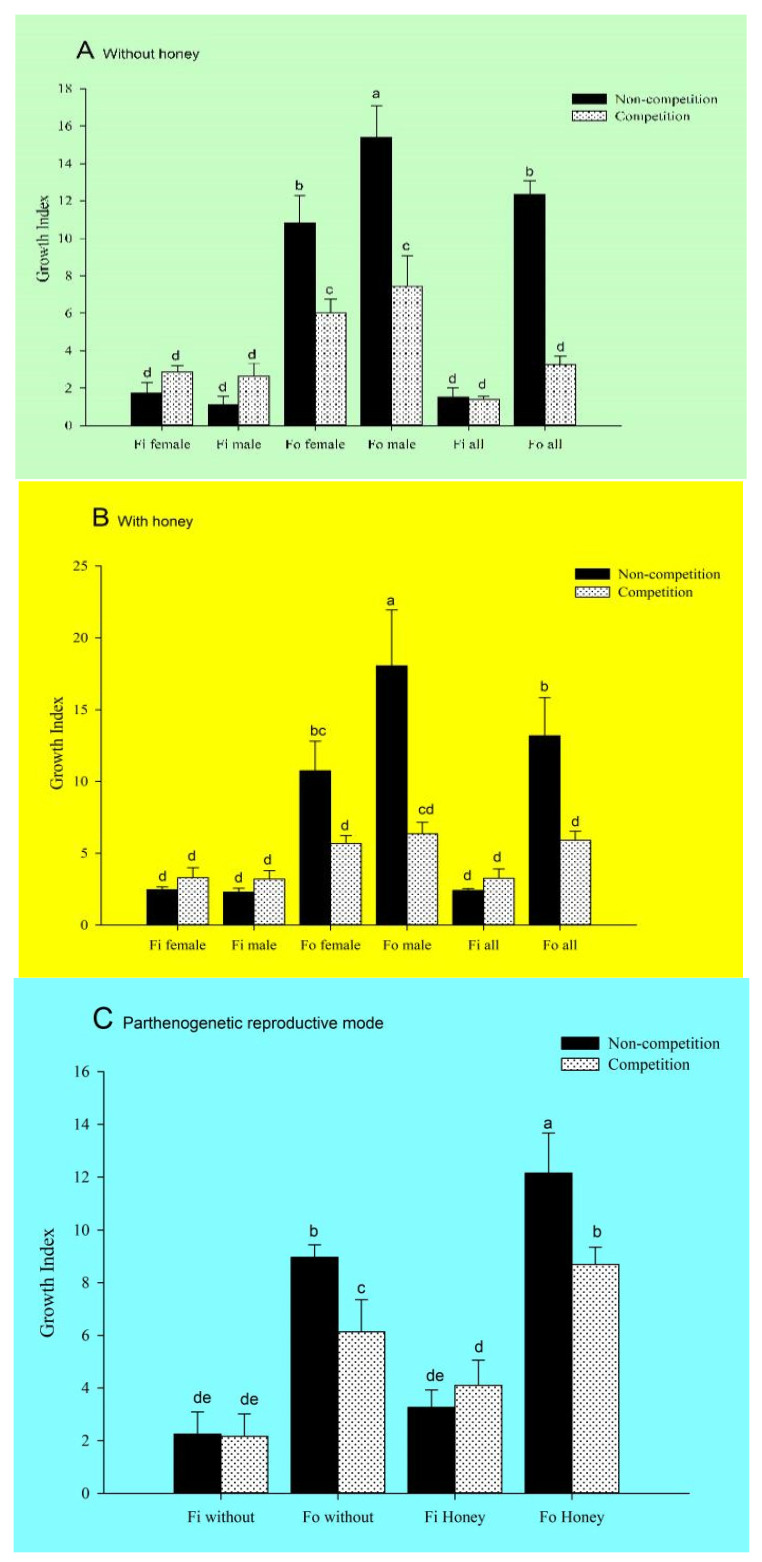
(**A**) The effect of interspecific competition between *F. occidentalis* (*Fo*) and *F. intonsa* (*Fi*), cultured under sexual reproductive mode without honey, on the growth index of adults (*F*_11,36_ = 27.84, *p* < 0.001). (**B**) The effect of interspecific competition between *Fo* and Fi, cultured under sexual reproductive mode with honey, on the growth index of adults (*F*_11,36_ = 14.98, *p* < 0.001). (**C**) The effect of interspecific competitions on the population growth index of *F. occidentalis* (*Fo*) and *F. intonsa* (*Fi*) cultured together under parthenogenetic reproductive mode with or without honey (*F*_7,16_ = 44.38, *p* < 0.001). The values are represented as the mean ± SEM. The letters above the columns indicate the comparison among groups, *p* < 0.05.

**Figure 2 insects-14-00001-f002:**
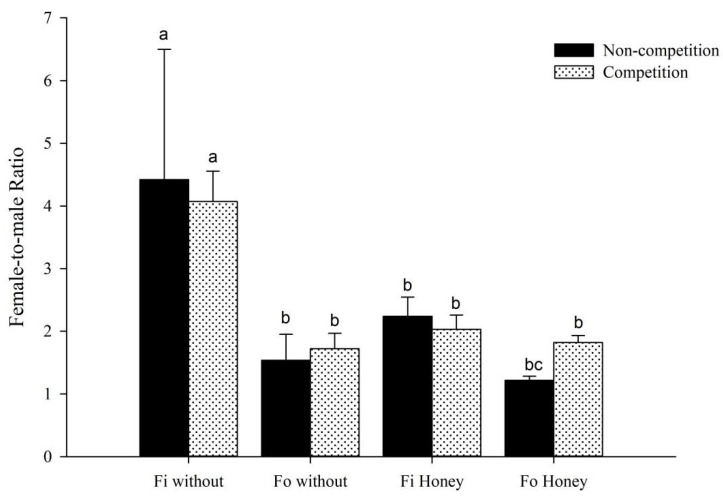
The effect of interspecific competitions on the female-to-male ratio of progeny of *F. occidentalis* (*Fo*) and *F. intonsa* (*Fi*) cultured together under sexual reproductive mode with or without honey (*F*_7,24_ = 1.57, *p* = 0.1916). The values are represented as the mean ± SEM. The letters above the columns indicate the comparison among groups, *p* < 0.05.

**Figure 3 insects-14-00001-f003:**
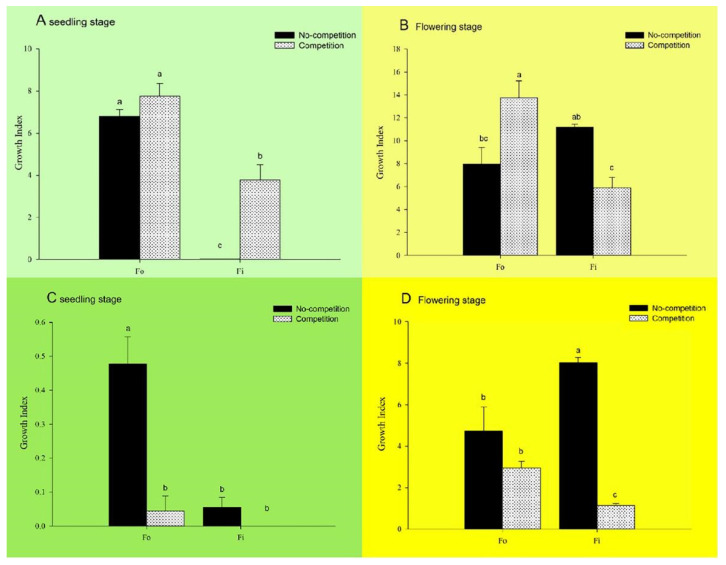
(**A**) The growth index of *F. occidentalis* (*Fo*) and *F. intonsa* (*Fi*) in non-competition population versus competition population under sexual reproduction mode feeding on pepper at the seedling stage (*F*_3,8_ = 148.98, *p* < 0.001). (**B**) The growth index of *Fo* and *Fi* in a non-competition population versus competition population under sexual reproduction mode feeding on pepper at the flowering stage (*F*_3,8_ = 9.11, *p* = 0.0058). (**C**) The growth index of *Fo* and *Fi* innon-competition population versus competition population under parthenogenetic reproduction mode feeding on pepper at the seedling stage (*F*_3,12_ = 11.33, *p* = 0.0008). (**D**) The growth index of *Fo* and *Fi* in non-competition population versus competition population under parthenogenetic reproduction mode feeding on pepper at the flowering stage (*F*_3,8_ = 22.57, *p* = 0.0003). The values are represented as the mean ± SEM. The letters above the columns indicate the comparison among groups, *p* < 0.05.

**Table 1 insects-14-00001-t001:** Experimental design.

No.	Competition	Number of *Fo* Females	Number of *Fo* Males	Number of *Fi* Females	Number of *Fi* Males	Food	Sexual Reproduction	Number of Replications
1	No	20	10	0	0	lentil with honey	Yes	4
	No	20	10	0	0	lentil without honey	Yes	4
	No	0	0	20	10	lentil with honey	Yes	4
	No	0	0	20	10	lentil without honey	Yes	4
	Yes	10	5	10	5	lentil with honey	Yes	4
	Yes	10	5	10	5	lentil without honey	Yes	4
2	No	20	0	0	0	lentil with honey	No	4
	No	20	0	0	0	lentil without honey	No	3
	No	0	0	20	0	lentil with honey	No	3
	No	0	0	20	0	lentil without honey	No	3
	Yes	10	0	10	0	lentil with honey	No	3
	Yes	10	0	10	0	lentil without honey	No	3
3	No	20	10	0	0	pepper seedlings stages	Yes	3
	No	20	10	0	0	pepper flowering stages	Yes	3
	No	0	0	20	10	pepper seedlings stages	Yes	3
	No	0	0	20	10	pepper flowering stages	Yes	3
	Yes	10	5	10	5	pepper seedlings stages	Yes	3
	Yes	10	5	10	5	pepper flowering stages	Yes	3
4	No	30	0	0	0	pepper seedlings stages	No	3
	No	30	0	0	0	pepper flowering stages	No	3
	No	0	0	30	0	pepper seedlings stages	No	3
	No	0	0	30	0	pepper flowering stages	No	3
	Yes	15	0	15	0	pepper seedlings stages	No	3
	Yes	15	0	15	0	pepper flowering stages	No	3

## Data Availability

The data presented in this study are available on request from the corresponding author.

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
