# Peer review of "Interspecific Competitions between Frankliniella intonsa and Frankliniella occidentalis on Fresh Lentil Bean Pods and Pepper Plants"

_insects, 2022, doi:10.3390/insects14010001_

Round 1
Reviewer 1 Report (Previous Reviewer 1)

Author Response
Please see the attachment

Reviewer 2 Report (Previous Reviewer 3)

Author Response
Please see the attachment.

This manuscript is a resubmission of an earlier submission. The following is a list of the peer review reports and author responses from that submission.
Round 1
Reviewer 1 Report
I recommend the publication: it only need few correction and add few missing data (please see the attached file of comments).

Author Response
General comments:
The manuscript reports on the clarification of the interspecific interactions between two thrips species and the evaluation of the effect of interspecific interactions on the population development of Fiand Fo under the bisexual and parthenogenetic reproductive modes in two hosts(isolated lentils with/without honey and pepper plants at seedling/flowering stages).
The topic addressed in in this study is extremely important for the knowledge of the reproductive biology and evolution of the species and also for the applied scope.The study carried out allowed the evolution of knowledge about the reproductive biology of the species,in particular,the intraspecific relationships that may occur mediated by the type of food,as well as inferring about the process of invasion and colonization of the habitat. Due to the importance of the practical implications in the rational management of these species,this aspect could be more emphasized in the conclusions.
The experiment was well designed, well executed and carefully analysed. In the manuscript,the objectives are well defined according to the experimental device presented.In addition to what has been achieved a very good framework literature on the subject and presented a novel approach methodology concerning the hormonal regulation expressed after mating.
For all these reasons,I recommend the publication;it only needs few corrections and add few missing data.
Specific comments:
Introduction
- On lines 55,56and 61,replace by Frankliniella
Response: Done.
Thanks for the suggestion.
I have corrected this according to your suggestion.
Materials and Methods
- On line 88,replace F.by Frankliniella
Response: Done.
Thanks for the suggestion.
I have corrected this according to your suggestion.
2)On lines 94-96,clarify how it was guaranteed that occur bisexual reproduction since with the adoption of this type of procedure, parthenogenetic reproduction can also coexist
Response: Done.
What you said is very good question. As for the insects that have the ability of 'bisexual reproduction' and 'parthenogenetic reproduction', even the males and females coexist in the same space, both 'bisexual reproduction' and 'parthenogenetic reproduction' still occur at the same time. Therefore, we can not show the absolute 'bisexual reproduction' status. The 'bisexual reproduction' in our manuscript represents the males and the females coexist in the same space.
Results
No sugestions
Discussion
1) On the conclusion lines 379-386,the authors may emphasized the practical implications of the results obtained, in terms of its importance to the rational management of these thrips species
Response: Done.
Thanks for the suggestion.
I have corrected this according to your suggestion.
References
Should the references not be presented in alphabetical order?
Response: Done.
Thanks for the suggestion.
References are arranged according to the order in which they appeared in the article.
Reviewer 2 Report
This is an interesting comparison of two thrips species, one of which is an invasive pest of global concern. There are many places were clarification is needed. Much of this is to clarify English. One term in particular stands out - a fluke. I'm not sure what that is in this context. Another key point - an important part of the core of this study is comparing parthenogenic versus bisexual modes. The basis of this must go into the introduction in order to understand the paper. You wait until the end to mention haplodipoidy, and even then, only for one species. Detailed comments follow:
Line 58 – better to say high fecundity.
Line 60 – feed in cryptic sites. They can cause injury [damage is more of an economic issue. Injury is better used for actual effects/manifestation of feeding behavior]
Line 61 – injure the leaves
Line 64 and throughout – injury rather than damage.
Lines 66-67 – first sentence is paragraph is incomplete, needs reworking.
Line 67 – remove Such as
Line 76 – compared with pepper plants.
Line 79 – cultured in a soil substrate.
Line 83 – per flowerpot. [understood to be one]
Line 88 – avoid starting a paragraph with an abbreviation.
Line 93 – explain single-headed larvae. It will be very helpful to explain the mechanism of parthenogenetic versus bisexual lines.
Line 118, 138 – replicated three times [omit for]
Line 124 – period after (Fig 2).
Line 132, 139, 148 – what is a fluke? Is this the right word?
Lines 194-196 – why not give more visibility to the significant difference, before listing the non-significant differences?
Fig 8A – the background color makes it hard to read the legends. Stay with consistent terminology. Interactive versus mixed populations.
Line 310 – here you bring up haplodiploidy, this needs to be done early – it’s key to understanding the research approach.
Line 375 – the two thrips. In nature….
Line 400 – spell out anonymous
Bibliography – there is a change in style between references 9 and 10. Also, inconsistent style regarding capitalization of words in article titles. Reference 8 – need to capitalize order and family. Reference 18 – is Thrips in the journal title an abbreviation? It probably doesn’t take a period.
Author Response
This is an interesting comparison of two thrips species, one of which is an invasive pest of global concern. There are many places were clarification is needed. Much of this is to clarify English. One term in particular stands out - a fluke. I'm not sure what that is in this context. Another key point - an important part of the core of this study is comparing parthenogenic versus bisexual modes. The basis of this must go into the introduction in order to understand the paper. You wait until the end to mention haplodipoidy, and even then, only for one species. Detailed comments follow:
- Line 58 – better to say high fecundity.
Response: Done.
Thanks for the suggestion.
We have corrected the spell.
- Line 60 – feed in cryptic sites. They can cause injury [damage is more of an economic issue. Injury is better used for actual effects/manifestation of feeding behavior]
Response: Done.
Thanks for the suggestion.
We have changed “damages” to “injury” according to your suggest.
- Line 61 – injure the leaves
Response: Done.
Thanks for the suggestion.
We have revised the original text according to your suggestion.
- Line 64 and throughout – injury rather than damage.
Response: Done.
Thanks for the suggestion.
We have revised the original text according to your suggestion.
- Lines 66-67 – first sentence is paragraph is incomplete, needs reworking.
Response: Done.
Thanks for the suggestion.
We have rewritten the first sentence of this part.
- Line 67 – remove Such as
Response: Done.
Thanks for the suggestion.
We have revised the original text according to your suggestion.
- Line 76 – compared with pepper plants.
Response: Done.
Thanks for the suggestion.
I have corrected this according to your suggestion.
- Line 79 – cultured in a soil substrate.
Response: Done.
Thanks for the suggestion.
I rewrote this paragraph.
- Line 83 – per flowerpot. [understood to be one]
Response: Done.
Thanks for the suggestion.
I rewrote this paragraph.
- Line 88 – avoid starting a paragraph with an abbreviation.
Response: Done.
Thanks for the suggestion.
I have corrected this in the text.
- Line 93 – explain single-headed larvae. It will be very helpful to explain the mechanism of parthenogenetic versus bisexual lines.
Response: Done.
Thanks for the suggestion.
I have changed the original text to: “Parthenogenetic rearing of F. occidentalis and F. intonsa was achieved by culturing the single larva in a 50 ml centrifuge tube with fresh lentil bean pods as food.”
- Line 118, 138 – replicated three times [omit for]
Response: Done.
Thanks for the suggestion.
I have checked the whole article and corrected the same mistakes.
- Line 124 – period after (Fig 2).
Response: Done.
Thanks for the suggestion.
I have corrected this according to your suggestion.
- Line 132, 139, 148 – what is a fluke? Is this the right word?
Response: Done.
Thanks for the suggestion.
“Fluke” is not proper here. I have corrected this according to your suggestion.
- Lines 194-196 – why not give more visibility to the significant difference, before listing the non-significant differences?
Response: Done.
Thanks for the suggestion.
I have put the significant differences before listing the non-significant differences in the MS.
- Fig 8A – the background color makes it hard to read the legends. Stay with consistent terminology. Interactive versus mixed populations.
Response: Done.
Thanks for the suggestion.
I have changed the background to a brighter color, and changed “or” to “versus”.
- Line 310 – here you bring up haplodiploidy, this needs to be done early – it’s key to understanding the research approach.
Response: Done.
Thanks for the suggestion.
I have added the explaination the of haplodiploidy in introduction. Details are as follows:
“Both F. intonsa and F. occidentalis are capable of sexual and parthenogenetic reproduction; in parthenogenetic reproduction, females do not need to mate to reproduce as unfertilised eggs develop into haploid males [35]. When both male and female thrips are present, they will produce diploid females from fertilized eggs and haploid males from unfertilised eggs [14]. A female-biased sex ratio was initially produced by all mated females [36]. “
- Line 375 – the two thrips. In nature….
Response: Done.
Thanks for the suggestion.
I have corrected this according to your suggestion.
- Line 400 – spell out anonymous
Response: Done.
Thanks for the suggestion.
I have corrected this according to your suggestion.
Reviewer 3 Report
The manuscript by Yang et al repots on a study on two species of flower thrips, the native Frankliniella intonsa (Fi) and the introduced F. occidentalis (Fo). They compare the growth index of sexual and parthenogenic thrips populations reared in the presence and absence of interspecific interactions and using different types of food (lentils with and without honey, pepper plants grown in pots). Studying the effect of flower thrips is important because they can impact the yield of economically valuable crops. The importance of studying thrips could have been emphasized by providing estimated of economic losses caused by F. intona and F. occidentalis.
The introduction would have benefitted from a brief description of the novely of this study, i.e., what relevant information is known, what relevant gaps in the literature exist, and how this study contributes to closing those gaps in knowledge. Both thrips species co-occur in the field, so it makes sense to study the effect of competition/facilitation on thrips performance. However, the introduction provides no information on why it is important (1) to study the performance of sexual and asexual morphs separately, (2) to know thrips performance on dried lentils with and without honey supplement, and (3) to study thrips population on live pepper plants in the seedling and flowering stage. The manuscript provides no hypothesis outlining the expected results of the experiments and underlying reasons for those expectations. It might have been better to use fewer treatments in favor of increasing the number of replications because each replicate provides one data point for calculating growth index and sex ratio. So, the results of each treatment consist of only 3-4 data points.
The Material and Methods section lacks important details about the study:
· What was the source for the lentils
· What fertilizer (M:P:K) was used to grow the pepper plants and how much and how frequently was it added to the plants.
· How big were the holes in the 12-hole trays
· Were the plants grown in a greenhouse, what were the rearing conditions?
· The lentils with honey were soaked in 10% honey water, were the lentils without honey soaked in water?
· A description of how you distinguish between the thrips species and between males and females?
The description of the experimental design could be shortened. The authors could provide a description of the methods that is the same for two or more treatments and then only describe the what the differences are. Figure 1-4 could be replaced by a table with the following headings:
Interaction (yes/no), Number of Fo females, number of Fo males, number of Fi females, number of Fi males, food (lentil with honey, lentil without honey, pepper seedlings, flowering pepper plants), sexual reproduction (yes/no), number of replications
Fig. 4 indicated lentils as food but according to the figure legend thrips were reared on pepper plants.
The data analysis states that: “The difference of various parameters before and after interactions was analyzed by the t test”. However, the t-test was to other comparisons as well. It seems that the authors did not account for multiple comparisons (growth index between non-interaction and interaction treatment of males and females, and growth index between species)
It is not explained how the growth index is calculated.
Results:
Figures: It is more conventional to indicate significant results with stars than not significant comparisons with bars. The abbreviations Fo and Fi are not defined anywhere.
It is very difficult to follow the description of the results because it reads like a list of bullet points without bullets. One possibility to improve the readability of the results might be to include all the results in a table including the statistics (df=value, p-value). Then in the description the authors could focus on highlighting the significant results. E.g., “In the lentil treatment we found that independent of honey supplementation the presence of F. intonsa had a significant negative effect on the growth index of male and female F. occidentalis.”
Discussion:
The discussion mostly repeats the results. The few explanations of the results are not sufficiently supported by the literature (the authors include only three different studies in their discussion). For instance, lines 308-317: The availability of honey increases the proportion of F. occidentalis females increased. It is not explained why a higher mating efficiency could explain the observed change in the sex ratio.
Lines 379-380: The study does not explore the effect of environmental factors on thrips populations, hence, the authors cannot conclude that “the outcomes of interspecific interactions between the two species of thrips are affected by many environmental factors”.
Overall, the manuscript is written in poor English. Some words do not make sense, e.g., what is the meaning of fluke in like 139? Spawning is not used for terrestrial animal egg laying.
